

# Citrus flavonoids, β-Glucan and organic acid feed additives decrease relative risk during *Yersinia ruckeri* O1 biotype 2 infection of rainbow trout (*Oncorhynchus mykiss*)

Kasper Rømer Villumsen[1,*], Maki Ohtani[1,*], Torunn Forberg[2], John Tinsley[3], Mette Boye[1] and Anders M. Bojesen[1]

[1] Preventive Veterinary Microbiology, Department of Veterinary and Animal Sciences , University of Copenhagen, Denmark, Frederiksberg, Denmark
[2] BioMar Group, Trondheim, Norway
[3] BioMar Group, Guayaquil, Ecuador
[*] These authors contributed equally to this work.

Corresponding authors
Kasper Rømer Villumsen,
krv@sund.ku.dk
Anders M. Bojesen, miki@sund.ku.dk

## ABSTRACT

Whether through direct supplementation of bacteria or by prebiotic supplementation thought to favour subsets of bacteria, modulation of gut microbiota constitutes an important and promising alternative to the use of prophylactic and growth promoting antibiotics in worldwide aquaculture. We fed a commercial base feed, alone or supplemented with either proprietary β-glucan, β-glucan and organic acids, citrus flavonoid or yeast cell wall supplements, to rainbow trout over a period of four weeks. Fish from each feed group were then subjected to experimental, waterborne infection with *Yersinia ruckeri* O1 biotype 2. Following experimental feeding, the β-glucan and organic acids supplemented group showed significantly improved feed conversion and lipid efficiency ratios. Furthermore, the β-glucan, β-glucan and organic acids and citrus flavonoid supplements proved to significantly reduce the risk of mortality in rainbow trout during experimental infection as shown by hazard ratio analysis. Resulting in 33.2%, 30.6% and 30.5% reduction in risk relative to the non-supplemented base feed, respectively, these three supplements show a promising potential either as stand-alone feed supplements, or as components in complex feed formulations.

## INTRODUCTION

Intensive rearing practices and continual growth in modern aquaculture places increasing demands on fish feed sourcing, management practices and disease prevention. Along with prophylactic measures such as vaccines, fish feed remains an important field of continual development. As the marine content has been drastically reduced to accommodate the increased demand (*Ytrestøyl, Aas & Åsgård, 2015*) and legislative action has been taken against antimicrobial growth promoters, attention has been focused at pre-, pro- and

synbiotic feed additives for further optimization (*Hoseinifar et al., 2017*; *Lauzon et al., 2014*; *Merrifield et al., 2010*). While prebiotics are compounds, often oligosaccharides, that are administered in order to promote a beneficial microbial composition (*Gibson et al., 2017*; *Lauzon et al., 2014*), probiotics are beneficial bacteria that are directly administered (*FAO & WHO, 2001*; *Hill et al., 2014*). Finally, synbiotic feed supplements incorporate both pre- and probiotics (*Gibson & Roberfroid, 1995*). By modulating the composition of the gut microbiota, the aims are to improve feed performance and feed component utilization, but also to improve disease resistance, whether through direct antagonism (*Wanka et al., 2018*) or competition (*Balcázar et al., 2008*).

Numerous types of substances have been tested in various aquaculture relevant species, for prebiotic as well as potentially immunomodulating properties, as recently reviewed by *Dawood, Koshio & Esteban (2018)*. These include yeast cell wall components, typically in the form of various preparations of β-glucans and mannan oligosaccharides (MOS) (*Skov et al., 2012*; *Staykov et al., 2007*; *Torrecillas, Montero & Izquierdo, 2014*), organic acids and their salts (*Gao et al., 2011*; *Hernández, Satoh & Kiron, 2012*; *Yilmaz & Ergun, 2018*; *Yılmaz et al., 2019*) and citrus flavonoids. Comprising more than one hundred compounds with antioxidant properties citrus flavonoids are found in *Citrus* fruits, particular in the peel (*Tripoli et al., 2007*; *Wang et al., 2017*), and have previously been shown to improve resistance towards bacterial infection by intraperitoneal injection in rainbow trout (*Acar et al., 2018*).

Previous studies using these feed supplements in various fish species have focused on mitigating adverse effects from antinutritional factors present in certain plant raw materials, growth performance and feed utilization, in some cases followed by experimental or natural infection with either bacterial or parasitic pathogens (*Acar et al., 2018*; *Gao et al., 2011*; *Hernández, Satoh & Kiron, 2012*; *Jaafar et al., 2013*; *Pandey & Satoh, 2008*; *Refstie et al., 2010*; *Staykov et al., 2007*; *Yilmaz & Ergun, 2018*; *Yılmaz et al., 2018*; *Yu et al., 2014*).

Given the importance of the gut homeostasis during bacterial infection (*Khimmakthong et al., 2013*; *Ohtani et al., 2014*; *Tobback et al., 2010*), as well as the hypothesised effects of a modulated gut microbiota, the aim of the present study was to identify prebiotic supplements that will function as stand-alone supplements and/or form the basis of synbiotic feed supplements guided by two focal points: feed performance and resistance towards experimental bacterial infection. An unsupplemented base feed, as well as proprietary experimental feeds supplemented with β-glucan, β-glucan combined with organic acids, citrus flavonoids or yeast cell wall extracts, either alone or in combination, was fed to rainbow trout fingerlings (*Oncorhynchus mykiss*). Growth and feed performance parameters were recorded, and finally a waterborne *Yersinia ruckeri* serotype O1 biotype 2 infection model was employed (*Ohtani et al., 2019b*). Using an isolate from a Danish freshwater trout farm outbreak, this infection model addresses all natural barriers of the fish, including the gastrointestinal tract, and represents a highly relevant pathogenic threat to commercial rainbow trout farming.

**Table 1  Composition of each feed group used in the experimental setup.**

| Group designation | Content |
| --- | --- |
| Control-Base feed | 48% protein, 23% fat, 20 MJ/kg, 35% marine content, 26% fishmeal |
| β-glucan | Base feed supplemented with β-glucans, vitamins C + E and nucleotides |
| β-glucan + OA | Base feed supplemented with β-glucans and organic acids, vitamins C + E and nucleotides |
| CF | Base feed supplemented with citrus flavonoids, vitamins C + E and nucleotides |
| YCW | Base feed supplemented with yeast cell wall extract vitamins C + E and nucleotides |

## MATERIALS & METHODS

### Ethics statement

The Danish Animal Experiments Inspectorate, under license no. 2015-15-0201-00645 approved the protocols regarding experimental animals described for this study. The study is thus approved under the Danish law regarding experimental animals.

### Rainbow trout

Rainbow trout eggs were acquired from AquaSearch Ova (AquaSearch FRESH, Billund, Denmark, 100% females). Following disinfection with Desamar K30 (Foodtech AG, Uster, Schweiz) according to the manufacturer's instructions, the eggs were hatched and reared under pathogen-free conditions at the Bornholm Salmon Hatchery (Nexø, Denmark). During the rearing period, the fish were fed commercial pelleted feed.

### Experimental feeding

Prior to experimental feeding, the fish were transported to the BioMar Research facility (Hirtshals, Denmark), divided into a total of 30 tanks with a volume of 150 l, each holding 29–32 individuals and allowed to acclimatize for 14 days. Once acclimated, each tank was assigned to one of five experimental feed groups resulting in 6 tanks/dietary group. Prior to the initiation of the experimental work, each feed group was anonymized, and the experiment was performed as a single-blinded setup. This facility has previously been described in detail (*Ohtani et al., 2019b*). In brief, all tanks were supplied through a closed recirculating water system. The water temperature was 14−15 °C, and filtering was performed through physical filtering, as well as a biofilter. The water was oxygenated and passed through an UV-filter (595 $\mu$J/cm$^2$). At the start of the experimental feeding period, the fish had an average weight of 24.4 g (bulk weights divided by number of individuals for all tanks).

Each experimental feed type was based on the same proprietary base feed (included as control), and supplemented as described in Table 1 to produce group specific two mm pellets. As all supplemented feeds are proprietary, only general descriptions are given. All pellets were produced by BioMar A/S, and the feeding period was 37 days (1.5% bodyweight/day).

As a technical error affected waterflow and oxygen supply to one tank in the control feed group, this tank was eliminated from the experiment and downstream analysis.

## Experimental feed performance

The performance of each feed group was evaluated based on weight gain (WG), feed conversion ratio (FCR), specific growth rate (SGR), lipid efficiency ratio (LER and protein efficiency ratio (PER). The equations used are given below:

1)
$$WG = \frac{\left(Final\ biomass(g) - Initial\ biomass\ (g)\right)}{Initial\ biomass\ (g)} \times 100\%$$

2)
$$FCR = \frac{Total\ consumed\ feed\ (g)}{Final\ biomass\ (g) - Initial\ biomass(g) + Dead\ biomass\ (g)}$$

3)
$$SGR = \frac{(Ln\left(Final\ biomass(g)\right)) - Ln(Initial\ biomass(g))}{Growth\ period\ (days)} \times 100\%$$

4)
$$LER = \frac{Gained\ biomass\ (g)}{Ingested\ fat\ (g)}$$

5)
$$PER = \frac{Gained\ biomass\ (g)}{Ingested\ protein\ (g)}$$

All calculations are made based on tank-wise bulk weighing of total biomass prior to and following the experimental feeding period. Administered feed was recorded for each tank throughout the experimental period and ingested fat/protein is calculated based on total ingested feed and the content of each component in each feed type.

## Experimental infection

Following the initial feeding period, all individuals from four of the six tanks in each experimental feed group were subjected to waterborne experimental bath infection with *Y. ruckeri*. The individuals from the remaining two tanks were subjected to mock-infection. Experimental feeding continued throughout the infection period.

The bath infection was performed as previously described (*Ohtani et al., 2019b*). Briefly, cryopreserved *Y. ruckeri* serotype O1 biotype 2 (strain 07111224) was cultured on blood agar plates for 48 h. Single colonies were then used to inoculate Luria-Bertani broth, which was subsequently incubated at room temperature for 36 h. Prior to infection, the bacteria were harvested by centrifugation and resuspended in clean tank water. Finally, fish from a given tank destined for infection were transferred to a designated infection tank with clean tank water. The infection was then started by the addition of bacterial suspension to a final concentration of $7.5 \times 10^8$ CFU/ml. The fish were kept in their respective infection tanks for 3 h, after which they were returned to their holding tanks

and monitored closely for 22 days. Mock-infections were performed by transferring fish to separate mock-infection tanks holding clean tank water for 3 h, before returning them to their respective holding tanks, thus mimicking the handling of the infected fish. Experimental feeding continued throughout the infection period, however reduced to 1.1% biomass/day. During the experimental infection period, all fish that met specified humane endpoints (distinct signs of established disease such as loss of equilibrium, protrusion of either or both eyes, haemorrhages along fin bases) were considered moribund, netted and euthanized with an overdose of benzocaine in accordance with the experimental animal license. The terms "survival", and consequently "mortality", in the following text reflect the binary nature of survival analyses and should therefore be considered technical, rather than purely descriptive terms. Following euthanasia, a swab was made from the anterior kidney onto 5% blood agar plates to re-isolate the bacterial pathogen in order to satisfy Koch's postulates.

## Statistical analysis

Throughout the analyses in the present study, a 95% confidence level was applied using a threshold $\alpha$-level of 0.05 for rejection of the null-hypothesis that there is no difference between the groups in question. All statistical analyses were performed using GraphPad Prism® for Mac (GraphPad, San Diego, USA).

Performance data were tested for underlying Gaussian distribution using the Kolmogorov–Smirnov test with Dallal-Wilkinson-Lilliefor correction. Having confirmed this, the means of each experimental group were compared to that of the control feed group using a one-way ANOVA followed by Dunnett's post hoc test multiple comparisons test.

Data from the experimental infection were analysed using the Kaplan–Meier survival analysis tool. Moribund fish from which the bacterial pathogen could be re-isolated were computed as mortalities. In case the pathogen could not be re-isolated, the individual was computed as a censored individual. Based on the course of the survival curves, the log-rank (Mantel-Cox) method was chosen for the analysis. Hazard ratios (log-rank method) were calculated pairwise between all experimental groups, as well. Relative percent survival (RPS) was calculated as follows:

6)

$$RPS = (1 - \frac{Mortality\,(supplemented\,group)}{Mortality\,(control\,group)}) \times 100\%$$

The Bonferroni correction for multiple comparisons was used when comparing individual survival curves, adjusting the $\alpha$-level by dividing it with the number of comparisons made within the framework of the dataset.

# RESULTS

## Experimental feed performance

Feed performance was determined following the experimental feeding period, and the results are shown in Table 2.

**Table 2   Weight gain (WG), feed conversion ratio (FCR), specific growth rate (SGR), lipid efficiency ratio (LER) and protein efficiency ratio (PER) data (mean ± standard deviation) for the experimental feeding period.** All calculations were performed as described above.

|  | Control | β-glucan | β-glucan + OA | CF | YCW |
|---|---|---|---|---|---|
| WG | 116.8 (±3.2) | 120.4 (±7.1) | 119.1 (±3.3) | 119.2 (±3.6) | 114.9 (±8.6) |
| FCR | 0.68 (±0.04) | 0.66 (±0.03) | 0.60 (±0.01)* | 0.63 (±0.02) | 0.64 (±0.07) |
| SGR (%) | 2.11 (±0.03) | 2.16 (±0.11) | 2.14 (±0.05) | 2.14 (±0.07) | 2.08 (±0.12) |
| LER | 6.57 (±0.37) | 6.91 (±0.29) | 7.46 (±0.16)* | 7.04 (±0.22) | 7.18 (±0.84) |
| PER | 3.10 (±0.17) | 3.23 (±0.13) | 3.41 (±0.07) | 3.26 (±0.10) | 3.30 (±0.39) |

**Notes.**

*Asterisks denote significant difference from control group values ($P < 0.05$).

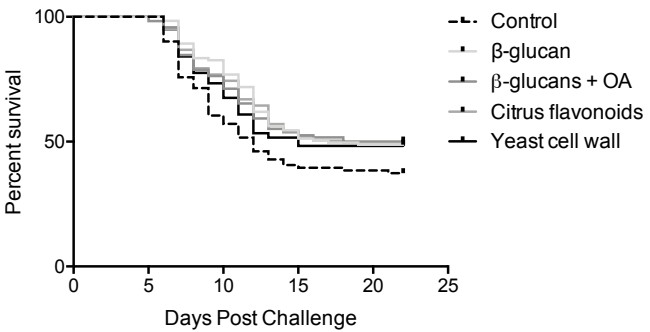

**Figure 1   Kaplan-Meier survival curves.** Kaplan-Meier survival curve analysis (Log-rank) of pooled replicates for each feed group. Statistical analysis is explained in Materials & Methods.

No differences were observed between the control group and any of the experimental groups regarding WG, SGR or PER. The group receiving the β-glucan + OA supplement was found to have a statistically significantly lower FCR, as well as a higher LER relative to the control group ($P < 0.05$).

## Experimental infection

Mortalities were observed in all infected tanks between 5- and 21-days post infection (Fig. 1). Log-rank survival curve analysis did not identify any statistically significant differences between replicate tanks in any of the groups ($P > 0.05$, Fig. S1). Consequently, data from replicate tanks were pooled for each group for all downstream analyses. The results from these analyses are summarized in Fig. 1 and Tables 2 and 3. Group-specific survival curves with 95% confidence intervals are shown in Fig. S2.

A single mortality occurred from one of the mock-infected tanks during the course of the experimental infection. However, given that only one occurred in a total of six tanks, this is considered negligible.

An initial, overall Log-rank survival curve analysis comparing all five survival curves at once did not identify any statistically significant differences between feed groups ($P = 0.13$). Pairwise comparisons between individual survival curves reached the same overall result, although offering a more nuanced view, as shown in Table 3. While the β-glucan, β-glucan + OA and CF supplemented groups all post $P$-values <0.05 when compared to the control

**Table 3** *P-values from pairwise log-rank survival curve analyses.* The Bonferroni-corrected α-level is 0.005.

| | Control | β-glucan | β-glucan + OA | CF | YCW |
|---|---|---|---|---|---|
| Control | – | 0.021 | 0.039 | 0.038 | 0.085 |
| β-glucan | | – | 0.886 | 0.890 | 0.567 |
| β-glucan + OA | | | – | 0.987 | 0.705 |
| CF | | | | – | 0.712 |
| YCW | | | | | – |

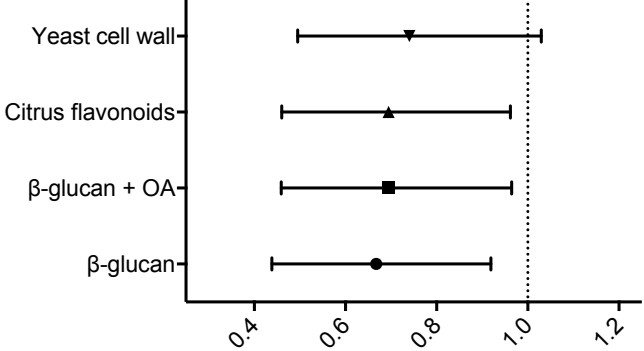

**Figure 2** **Hazard ratios relative to the control group.** Hazard ratios relative to the control group, including 95% CI. The vertical dotted line represents a hazard ratio of 1, indicating no difference in relative risk. Statistical analysis is explained in Materials & Methods.

feed group, correction for multiple comparisons reduces the α-level to 0.005. Their respective survival curves are therefore not significantly different from the control feed group. Comparisons among the supplemented groups resulted in high *P*-values (>0.5) indicating similar survival curves in these groups.

When subsequently comparing the relative risk of mortality following infection using the hazard ratios comparing the control group and each of the experimental feed groups in turn, all but the yeast cell wall supplemented feed group displayed hazard ratios with 95% confidence intervals <1 (see Fig. 2 and Table 4) These indicate statistically significant reductions in risk following infection relative to the control group (*Motulsky, 2014*).

## DISCUSSION

Following the experimental feeding period, the β-glucan + OA supplemented feed group showed signs of improved feed utilization when compared to the control feed group, as well as the β-glucan supplemented feed group. As the study design did not include a group supplemented with organic acids alone, we are unable to assess whether the growth parameters observed for the combined feed is due to the organic acid component or synergies between the β-glucan and organic acid components. This relationship should be investigated in future studies.

**Table 4  Results from the experimental infection.** All hazard ratios are relative to the control group. See Materials & Methods for details.

| Group | Endpoint survival | RPS | Hazard ratio (95% CI) |
|---|---|---|---|
| Control | 37.4% | – | – |
| β-glucan | 48.8% | 18.2% | 0.6677 (0.4385–0.9192) |
| β-glucan + OA | 50.0% | 20.2% | 0.6937 (0.4591–0.9647) |
| CF | 48.8% | 18.2% | 0.6946 (0.4605–0.9622) |
| YCW | 48.3% | 17.5% | 0.7400 (0.4952–1.030) |

The lowered FCR indicates a significantly increased capacity for converting ingested feed into biomass, while the increased LER suggests that fish in this group utilize the lipid content of the feed in a significantly more efficient manner. A significantly improved utilization of lipid content is noteworthy due to the current issues regarding sourcing of marine content for fish feed (*Ytrestøyl, Aas & Åsgård, 2015*). Previous studies have demonstrated a positive effect of either citric or amino acid supplements on retention of phosphorous, SGR and feed utilization in rainbow trout feed (*Hernández, Satoh & Kiron, 2012*). While Pandey & Satoh found that methionine hydroxy analogue supplementation of rainbow trout feed improved the FCR (*Pandey & Satoh, 2008*), a blend of sodium formate and butyrate was found to improve FCR from fishmeal-based rainbow trout feed (*Gao et al., 2011*). The same blend, however, was found to decrease the uptake of crude fat from plant-based feed and both effects were found to depend on the extrusion method used. Finally, *Yılmaz et al. (2018)* found that while 0.6% humic acid sodium salt supplementation improved digestive enzyme activities, this did not translate into changes in FCR or other growth parameters. The results from these studies suggests a level of complexity regarding the use of organic acid supplements, most likely reflecting the diversity of this class of supplements. As well as the exact nature of the supplement, dosage is also an important point. *Acar et al. (2018)* demonstrated a dose-dependent effect of pomegranate seed oil (a crude source of the organic punicic acid) supplemented feed on FCR and SGR in rainbow trout.

The results from the experimental *Y. ruckeri* infection indicated a substantial impact on the experimental groups, leading to a control group endpoint survival of 37.4%. The level of infection obtained is therefore considered sufficient, although rather severe, in terms of providing a realistic platform for evaluating the effects of the experimental prebiotic diets included in the study. Generally, waterborne infections of more moderate severity would be preferable when evaluating the effect of feed supplements on disease resistance, as stated in our previous studies (*Ohtani et al., 2019a*). While the initial, overall log-rank survival curve analysis did not identify statistically significant differences between the different feed groups, individual pairwise comparisons showed that the β-glucan, β-glucan + OA and CF group survival curves differed noticeably, however not statistically significantly, from that of the control group. When estimating the experiment-wide risks calculated for each experimental group relative to that of the control group, the resulting hazard ratios demonstrate statistically significantly reduced risk in the β-glucan, β-glucan + OA and CF fed groups, relative to that of the control group.

While reductions in relative risk observed in this study could be a consequence of the improved performance reported in the present study, the fact that β-glucans alone, as well as citrus flavonoid supplementation also resulted in significantly reduced relative risk during infection, without the improved feed performance, indicates a more complex background for the observed reductions. While increased fish size has been shown to result in reduced mortalities following bath infection (*Ohtani et al., 2019b*), the effects of this will be limited within the scope of this study, and supplement specific characteristics will thus play important roles.

The diet including β-glucan alone reduced the relative risk during infection by 33.2%, with a 95% CI that lies well below 1, and the combination of β-glucan and organic acids proved to reduce the risk of mortality by 30.6% relative to that of the control group, also with a 95% CI placed below 1. Previous study results on β-glucan supplemented feed in rainbow trout have diverged. *Skov et al. (2012)* found no effect of paramylon (1% w/w), a purified β-glucan product from *Euglena gracilis*, on rainbow trout resistance towards a waterborne experimental infection with *Y. ruckeri*. Meanwhile, Ji et al. reported significantly increased endpoint survival following a short intraperitoneal infection with *Aeromonas salmonicida* in rainbow trout fed 0.05–0.2% β-glucan from *Saccharomyces cerevisiae*. Besides a nearly 30-fold difference in fish size, these studies vary with respect to pathogen, infection model, statistical analysis and source of β-glucan, making them hard to compare directly. Our current study design does, however, lie closer to the former. As the potential impact of feed supplements is expected to be less than that of a traditional prophylactic measure, e.g., vaccination, the experimental infection model can prove highly important when assessing their effects (*Ohtani et al., 2019a*). This, as well as the use of hazard ratio analyses could account for the fact that the current study is able to demonstrate an effect of β-glucan on the risk of mortality during experimental infection. While the present study did not investigate immune mechanisms, Skov et al. observed an increased expression of lysozyme in β-glucan fed rainbow trout, however, with no significant increase in lysozyme activity (*Skov et al., 2012*). Furthermore, Schmitt et al. observed an increased number of cathelicidin producing cells in the proximal intestine of rainbow trout fed β-glucan (*Schmitt et al., 2015*), suggesting a role for these bactericidal peptides.

For organic acid supplements, an antimicrobial effect has been suggested through lowering of the pH of either the host intestinal environment or the cytoplasm of bacteria (*Ng & Koh, 2017*). This has recently been demonstrated for red hybrid tilapia (*Oreochromis* sp.) fed feed supplemented with a proprietary organic acid blend, where inclusion of this blend resulted in significant reduction in gut pH, fecal CFU counts, and significantly reduced mortalities following waterborne infection with *Streptococcus agalactiae* (*Koh et al., 2016*). In rainbow trout, studies on trans-cinnamic and humic acid supplements have been shown to result in significantly upregulated innate immune mechanisms including phagocytosis and lysozyme activity from blood samples, that translated into clear and statistically significantly improved survival following intraperitoneal infection with *Y. ruckeri* (*Yilmaz & Ergun, 2018*; *Yılmaz et al., 2018*). In addition to this, a recent study by Acar et al. investigated the effects of supplementation with pomegranate seed oil containing punicic acid in feed on resistance towards intraperitoneal *Y. ruckeri* infection in rainbow

trout, and found a higher, albeit not statistically significant, survival in supplemented groups relative to the control (*Acar et al., 2018*). The tendency evident from these previous studies support those from the present study. Bypassing mucosal barriers, including the gastrointestinal tract, this suggests a beneficial effect on systemic innate immune mechanisms. *Jaafar et al. (2013)* demonstrated that feed supplemented with a mix of propionic acid, formic acid and silicon dioxide resulted in a shift in allochthonous gut microbiota of rainbow trout, with no apparent effect on disease resistance following low-dose, long-term waterborne infection with *Y. ruckeri* serotype O1 biotype 2. In addition to this, feed co-supplementation with *Bacillus subtilis* and trans-cinnamic acid reportedly boosted the growth of *B. subtilis* in rainbow trout (*Yılmaz et al., 2019*). The potential for modulation of the gut microbiota, as well as the immune system using an organic acid supplement is therefore highly plausible. The co-supplementation with β-glucan and organic acids in this study, however, does not allow a distinction between their respective contributions.

The citrus flavonoid supplemented feed group showed a 30.5% reduction in relative risk with a 95% CI below 1. In a previous study Öntas et al. demonstrated the composition, and antibacterial effect of lemon peel oil against *Y. ruckeri* (*ÖntaŞ et al., 2016*). While this is a potent source of flavonoids (*Wang et al., 2017*), it remains a crude mix of multiple components. Specific bacteriostatic and antimicrobial effects of citrus flavonoids have been demonstrated from bergamot (*Citrus bergamia* Risso) peel extractions (*Mandalari et al., 2007*). Peel extractions, either native or pectinase treated, as well as purified flavonoids proved mainly effective against Gram-negative bacteria. A study by Vikram et al. indicates that purified citrus flavonoids can act as quorum sensing inhibitors, modulators of bacterial growth and inhibitors of type three secretion system component expression in *Vibrio harveyi* (*Vikram et al., 2010*). The presence of a type three secretion system in *Y. ruckeri* has recently been suggested by *Kumar et al. (2017)*, and a combination of general antimicrobial activities and inhibition of pathogenicity could thus be a plausible explanation for the observed reduction in pathogenicity. This would, however, require further studies into the exact molecular effects of the flavonoid supplement in question, and is therefore merely speculative for now.

The yeast cell wall supplement did not confer a significant reduction in relative risk in the present study. *Yu et al. (2014)* have previously shown that yeast cell wall extract, including 28% glucans and 24% mannan, improved resistance towards intramuscular infection with *Aeromonas veronii* in Japanese seabass (*Lateolabrax japonicus*). While mannan oligosaccharides supplementation has been associated with increased innate immune responsiveness (*Staykov et al., 2007*) and proposed to limit pathogen establishment through increased mucus production (*Torrecillas, Montero & Izquierdo, 2014*), and as β-glucan supplementation has proven successful in this study, as well as in previous studies, crude yeast cell wall extract fails to confer the same reduction in relative risk during infection in the present study.

## CONCLUSIONS

β-glucan + OA supplementation resulted in improved feed utilization relative to the base control. Furthermore, β-glucan, β-glucan and organic acids, as well as citrus flavonoid supplementation proved to significantly reduce the relative risk of mortality in rainbow trout during experimental infection with *Y. ruckeri* O1 biotype 2. These prebiotic and future derived synbiotic feed supplements can therefore be expected to positively affect feed performance and disease resistance in cultured rainbow trout.

## ACKNOWLEDGEMENTS

Anni Nielsen, Miguel Martin and the staff at BioMar Research Facility, Hirtshals, Denmark are acknowledged for their help and expertise throughout the experimental period.

### Funding

This study was funded by GUDP ("Præ-Pro-Fisk", grant no. 34009-17-1218) and BioMar A/S. The funders had no role in study design, data collection and analysis, decision to publish, or preparation of the manuscript.

### Grant Disclosures

The following grant information was disclosed by the authors:
GUDP ("Præ-Pro-Fisk": 34009-17-1218.
BioMar A/S.

### Competing Interests

Torunn Forberg and John Tinsley are both employed at different branches of Biomar Group, who produce, market and sell fish feed supplements with some of the ingredients tested in the current investigation. Furthermore, Biomar provided parts of the funding for this study.

### Author Contributions

- Kasper Rømer Villumsen analyzed the data, prepared figures and/or tables, authored or reviewed drafts of the paper, and approved the final draft.
- Maki Ohtani performed the experiments, analyzed the data, authored or reviewed drafts of the paper, and approved the final draft.
- Torunn Forberg conceived and designed the experiments, performed the experiments, analyzed the data, authored or reviewed drafts of the paper, and approved the final draft.
- John Tinsley and Mette Boye conceived and designed the experiments, authored or reviewed drafts of the paper, and approved the final draft.
- Anders M. Bojesen conceived and designed the experiments, analyzed the data, authored or reviewed drafts of the paper, and approved the final draft.

PeerJ ____________________________________________

## Animal Ethics

The following information was supplied relating to ethical approvals (i.e., approving body and any reference numbers):

The Danish Animal Experiments Inspectorate approved the protocols regarding experimental animals described for this study. The study is thus approved under the Danish law regarding experimental animals (no. 2015-15-0201-00645).

## Data Availability

The data are available in the Supplementary Files.

## Supplemental Information

Supplemental information for this article can be found online at http://dx.doi.org/10.7717/peerj.8706#supplemental-information.

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
