# Peer review of "Citrus flavonoids, β-Glucan and organic acid feed additives decrease relative risk during Yersinia ruckeri O1 biotype 2 infection of rainbow trout (Oncorhynchus mykiss)"

_PeerJ, doi:10.7717/peerj.8706_

## Round 0.1 · original submission · Major Revisions

The reviewers have commented on your above paper. They indicated that it is not acceptable for publication in its present form. The main concern is that some key papers available on this topic have not been considered in your manuscript. A more deep revision on the theme of the paper has to be done in order to improve it.

·

Basic reporting

Literature references, no sufficient field background/context provided.

Experimental design

No comment.

Validity of the findings

No comment.

Additional comments

Major comments:

Alot of work has been done in fish regarding b-glucan and organic acids effects on fish. Moreover, the experimental approach is very similar to a myriad of studies on all types of immunostimulants in fish. So, the authors should try to demonstrate that their approach gives an additional novelty to what has already been assessed in fish immune stimulation, otherwise it will be another one of these similar studies:

1-Skov, J., Kania, P. W., Holten-Andersen, L., Fouz, B., & Buchmann, K. (2012). Immunomodulatory effects of dietary β-1, 3-glucan from Euglena gracilis in rainbow trout (Oncorhynchus mykiss) immersion vaccinated against Yersinia ruckeri. Fish & shellfish immunology, 33(1), 111-120.
2- Yılmaz, S., & Ergün, S. (2018). Trans-cinnamic acid application for rainbow trout (Oncorhynchus mykiss): I. Effects on haematological, serum biochemical, non-specific immune and head kidney gene expression responses. Fish & shellfish immunology, 78, 140-157.
3- Yılmaz, S., Ergun, S., Çelik, E. Ş., & Yigit, M. (2018). Effects of dietary humic acid on growth performance, haemato‐immunological and physiological responses and resistance of Rainbow trout, Oncorhynchus mykiss to Yersinia ruckeri. Aquaculture research, 49(10), 3338-3349.
4-Yilmaz, S., Ergün, S., & Yıgıt, M. (2018). Effects of dietary FARMARIN® XP supplement on immunological responses and disease resistance of rainbow trout (Oncorhynchus mykiss). Aquaculture, 496, 211-220.
5- Yılmaz S, Ergun S, Yigit M, Çelik EŞ. (in press). Effect of combination of dietary Bacillus subtilis and transcinnamic acid on innate immune responses and resistance of rainbow trout, Oncorhynchus mykiss to Yersinia ruckeri. Aquac Res. 2019;00:1–14. https ://doi.org/10.1111/are.14379
6-Baba, E., Acar, Ü., Yılmaz, S., Zemheri, F., & Ergün, S. (2018). Dietary olive leaf (Olea europea L.) extract alters some immune gene expression levels and disease resistance to Yersinia ruckeri infection in rainbow trout Oncorhynchus mykiss. Fish & shellfish immunology, 79, 28-33.
7- Adel, M., Pourgholam, R., Zorriehzahra, J., & Ghiasi, M. (2016). Hemato–Immunological and biochemical parameters, skin antibacterial activity, and survival in rainbow trout (Oncorhynchus mykiss) following the diet supplemented with Mentha piperita against Yersinia ruckeri. Fish & shellfish immunology, 55, 267-273.
8- Acar, Ü., Parrino, V., Kesbiç, O. S., Lo Paro, G., Saoca, C., Abbate, F., ... & Fazio, F. (2018). Effects of different levels of pomegranate seed oil on some blood parameters and disease resistance against Yersinia ruckeri in rainbow trout. Frontiers in Physiology, 9, 596.
9- Dehghan, F., Vazirzadeh, A., Soltanian, S., Karami, A., & Akhlaghi, M. (2016). Mortality rate and immune responses of rainbow trout (Oncorhynchus mykiss) infected with Yersinia ruckeri subsequent to feeding on diet supplemented with Ducrosia anethifolia essential oil. International Journal of Aquatic Biology, 4(5), 340-344.
10-Şahan, A., Duman, S., Çolak, S. Ö., Çinar, E., & Bilgin, R. (2017). Determination of some hematological and non-specific immune defences, oxidative stress and histopathological status in rainbow trout (Oncorhynchus mykiss) fed rosehip (Rosa canina) to Yersinia ruckeri. Turkish Journal of Fisheries and Aquatic Sciences, 17(1), 91-100.
11-Zeraatpisheh, F., Firouzbakhsh, F., & Khalili, K. J. (2018). Effects of the macroalga Sargassum angustifolium hot water extract on hematological parameters and immune responses in rainbow trout (Oncohrynchus mykiss) infected with Yersinia rukeri. Journal of applied phycology, 30(3), 2029-2037.
12-Farsani, M. N., Hoseinifar, S. H., Rashidian, G., Farsani, H. G., Ashouri, G., & Van Doan, H. (2019). Dietary effects of Coriandrum sativum extract on growth performance, physiological and innate immune responses and resistance of rainbow trout (Oncorhynchus mykiss) against Yersinia ruckeri. Fish & shellfish immunology, 91, 233-240.
13-Bulfon, C., Bongiorno, T., Messina, M., Volpatti, D., Tibaldi, E., & Tulli, F. (2017). Effects of P anax ginseng extract in practical diets for rainbow trout (O ncorhynchus mykiss) on growth performance, immune response and resistance to Y ersinia ruckeri. Aquaculture research, 48(5), 2369-2379.

Should be detailed as "...................Various reports have been published on the use and effects of various dietary additives for O. mykiss exposed to Y. ruckeri pathogen. Among these addititives ............................................... are the most salient ones (1-13), with reference to resistance of rainbow trout against Y. ruckeri. However, different than the present study, none of these earlier reports presented .........................."

Line 270: You infected the fish with Yersinia ruckeri. Relevant references should be added and discussied. You can see in line 68 balloon.

Line 275-286: Where is the in vivo studies ? in vivo studies should be discussed.
You can see below:

Acar, Ü., Kesbiç, O. S., Yılmaz, S., Gültepe, N., & Türker, A. (2015). Evaluation of the effects of essential oil extracted from sweet orange peel (Citrus sinensis) on growth rate of tilapia (Oreochromis mossambicus) and possible disease resistance against Streptococcus iniae. Aquaculture, 437, 282-286.
Baba, E., Acar, Ü., Öntaş, C., Kesbiç, O. S., & Yılmaz, S. (2016). Evaluation of Citrus limon peels essential oil on growth performance, immune response of Mozambique tilapia Oreochromis mossambicus challenged with Edwardsiella tarda. Aquaculture, 465, 13-18.
Kesbiç, O. S., Acar, Ü., Yilmaz, S., & Aydin, Ö. D. (2019). Effects of bergamot (Citrus bergamia) peel oil-supplemented diets on growth performance, haematology and serum biochemical parameters of Nile tilapia (Oreochromis niloticus). Fish physiology and biochemistry, 1-8.
Lopes, J. M., de Freitas Souza, C., Saccol, E. M. H., Pavanato, M. A., Antoniazzi, A., Rovani, M. T., ... & Baldisserotto, B. (2019). Citrus x aurantium essential oil as feed additive improved growth performance, survival, metabolic, and oxidative parameters of silver catfish (Rhamdia quelen). Aquaculture nutrition, 25(2), 310-318.
Vicente, I. S., Fleuri, L. F., Carvalho, P. L., Guimarães, M. G., Naliato, R. F., Müller, H. D. C., ... & Barros, M. M. (2019). Orange peel fragment improves antioxidant capacity and haematological profile of Nile tilapia subjected to heat/dissolved oxygen‐induced stress. Aquaculture research, 50(1), 80-92.
Rahman, A. N. A., ElHady, M., & Shalaby, S. I. (2019). Efficacy of the dehydrated lemon peels on the immunity, enzymatic antioxidant capacity and growth of Nile tilapia (Oreochromis niloticus) and African catfish (Clarias gariepinus). Aquaculture, 505, 92-97....................................................................................................
You can see more study at https://scholar.google.com.tr/scholar?hl=tr&as_sdt=2005&sciodt=0%2C5&as_ylo=2019&cites=11700966083561767066&scipsc=&q=Citrus+peel+aquaculture&btnG=&oq=Citrus+peel+aquacu

Reviewer 2 ·

Basic reporting

Literature references, no sufficient field background/context provided.

Experimental design

Experimental designed well.

Validity of the findings

no comment

Additional comments

The introduction is relevant but must include new references. The discussion, in the light of results and knowledge, is relevant.
Acar, Ü., Parrino, V., Kesbiç, O. S., Lo Paro, G., Saoca, C., Abbate, F., ... & Fazio, F. (2018). Effects of different levels of pomegranate seed oil on some blood parameters and disease resistance against Yersinia ruckeri in rainbow trout. Frontiers in Physiology, 9, 596.

Baba, E., Acar, Ü., Yılmaz, S., Zemheri, F., & Ergün, S. (2018). Dietary olive leaf (Olea europea L.) extract alters some immune gene expression levels and disease resistance to Yersinia ruckeri infection in rainbow trout Oncorhynchus mykiss. Fish & shellfish immunology, 79, 28-33.

Acar, Ü., Kesbiç, O. S., Yılmaz, S., Gültepe, N., & Türker, A. (2015). Evaluation of the effects of essential oil extracted from sweet orange peel (Citrus sinensis) on growth rate of tilapia (Oreochromis mossambicus) and possible disease resistance against Streptococcus iniae. Aquaculture, 437, 282-286.
Baba, E., Acar, Ü., Öntaş, C., Kesbiç, O. S., & Yılmaz, S. (2016). Evaluation of Citrus limon peels essential oil on growth performance, immune response of Mozambique tilapia Oreochromis mossambicus challenged with Edwardsiella tarda. Aquaculture, 465, 13-18.
Kesbiç, O. S., Acar, Ü., Yilmaz, S., & Aydin, Ö. D. (2019). Effects of bergamot (Citrus bergamia) peel oil-supplemented diets on growth performance, haematology and serum biochemical parameters of Nile tilapia (Oreochromis niloticus). Fish physiology and biochemistry, 1-8.

Reviewer 3 ·

Basic reporting

'no comment'

Experimental design

'no comment'

Validity of the findings

'no comment'

Additional comments

This paper describes an investigation of the effects of feed additives on survival after experimental infection with Yersinia ruckeri. The paper is concise and to the point, the writing is clear, and the methods used are appropriate. In all, this is a nice piece of work that is of interest and potential use to the aquaculture industry.
I have just a few very minor questions/comments for the authors to consider.

Line 65: effect of four prebiotic feed supplements on fish growth….
Line 94: I don’t see a mention of what the water temperature(s) were for this study. This is an important parameter especially for the Yersinia ruckeri challenge portion of the study.
Line 97: (595 uj/cm) close brackets
Figure 1: I think you should plot the unchallenged control here. I see that this control was done.

·

Basic reporting

Generally this is an interesting work associated with the way to increase the survival of fish against Yersinia septicemia.

Experimental design

The authors preforemed their works in a good experiemntal design.

Validity of the findings

It is worth of furher processing of publication.

Additional comments

I think authors have a look at more recent data associated with the butyrate acid published in some journals such as Fish and Shellfish Immunology (2018 and 2019). It may worth to use such data for more validating of their data.

Reviewer 5 ·

Basic reporting

1. The article has not much data. The results part has too much explanation of the challenge result. The author should present data in a concise way.

2. I don't see information about the dose of the supplements. This should be clearly indicated in the article.

3. Normally, "Growth (%) " is presented as "weight gain (WG)"

Experimental design

The experimental design is OK.

Validity of the findings

The article doesn't include much data. Only growth performance and challenge data were included. Most of the data were negative. Although the difference between control and treatment was significant by hazard ratio analysis, this is not a typical way to analyze mortality data in fish experiment. The author should think about the dose and feeding duration.

Additional comments

This manuscript doesn't include enough solid and positive data for publication. The results can be presented in a much more concise way.

Reviewer 6 ·

Basic reporting

Kasper et al have shown that Citrus Flavonoids, beta-Glucan and Organic Acid Feed Additives Decrease Relative Risk During Yersinia ruckeri O1 Biotype 2 Infection of Rainbow Trout (Oncorhynchus mykiss).

Experimental design

However, 1,Why evaluate the use of these four feed additives, and what is the relationship among these additives. 2,The group receiving the beta-glucan + OA supplement was found to have a statistically significantly lower FCR, but the beta-glucan group. Therefore, it is necessary to study whether OA plays a major role. 3, No attempt to explain why additives enhance disease resistance.

Validity of the findings

no comment

Additional comments

no comment

---

## Round 0.2 · accepted · Accept

Thank you for considering all suggested corrections or additions in the new version of he paper. Now I am pleased to confirm that your paper has been accepted for publication in PeerJ.

Thank you also for submitting your work to this journal.

Reviewer 2 ·

Basic reporting

All suggested corrections or additions have been made by the authors.

Experimental design

Experimental designed is enough.

Validity of the findings

The findings of the study is sufficent

Reviewer 3 ·

Basic reporting

'no comment'

Experimental design

'no comment'

Validity of the findings

'no comment'

Additional comments

The authors have responded to my comments satisfactorily